# CAN TRANSFORMERS REPLACE CNNS FOR MEDICAL IMAGE CLASSIFICATION?

## ABSTRACT

Convolutional Neural Networks (CNNs) have reigned for a decade as the de facto approach to automated medical image diagnosis, pushing the state-of-the-art in classification, detection and segmentation tasks. Recently, vision transformers (ViTs) have appeared as a competitive alternative to CNNs, yielding impressive levels of performance in the natural image domain, while possessing several interesting properties that could prove beneficial for medical imaging tasks. In this work, we explore whether it is feasible to switch to transformer-based models for medical image classification as well, or if we should keep working with CNNs – can we trivially replace CNNs with transformers? We consider this question in a series of experiments on several standard medical image benchmark datasets and tasks. Our findings show that, while CNNs perform better if trained from scratch, off-the-shelf vision transformers can perform on par with CNNs when pretrained on ImageNet, both in a supervised and self-supervised setting.

## 1 INTRODUCTION

Breakthroughs in medical image analysis over the past decade have been largely fuelled by convolutional neural networks (CNNs). CNN architectures have served as the workhorse for numerous medical image analysis tasks including breast cancer detection (Wu et al., 2019), ultrasound diagnosis (Christiansen et al., 2021), and diabetic retinopathy (De Fauw et al., 2018). Whether applied directly as a plug-and-play solution or used as the backbone for a bespoke model, CNNs such as ResNet (He et al., 2016) are the dominant model for medical image analysis. Recently, however, vision transformers have gained increased popularity for natural image recognition tasks, possibly signalling a transition from convolution-based feature extractors to attention-based models (ViTs). This raises the question: can ViTs also replace CNNs in the medical imaging domain?

In the natural image domain, transformers have been shown to outperform CNNs on standard vision tasks such as IMAGENET classification (Dosovitskiy et al., 2020), as well as in object detection (Carion et al., 2020) and semantic segmentation (Ranftl et al., 2021). Interestingly, transformers succeed despite their lack of inductive biases tied to the convolution operation. This may be explained, in part, by the attention mechanism central to transformers which offers several key advantages over convolutions: it explicitly models and more efficiently captures long-range relationships (Raghu et al., 2021), and it has the capacity for adaptive modeling via dynamically computed self-attention weights that capture relationships between tokens. In addition, it provides a type of built-in saliency, giving insight as to what the model focused on (Caron et al., 2021).

Over the years, we have gained a solid understanding of how CNNs perform on medical images, and which techniques benefit them. For example, it is well-known that performance drops dramatically when training data is scarce (Cho et al., 2015). This problem is particularly acute in the medical imaging domain, where datasets are smaller and often accompanied by less reliable labels due to the inherent ambiguity of medical diagnosis. For CNNs, the standard solution is to employ transfer learning (Morid et al., 2020): typically, a model is pretrained on a larger dataset such as IMAGENET (Deng et al., 2009) and then fine-tuned for specific tasks using smaller, specialized datasets. For medical imaging, CNNs pre-trained on IMAGENET typically outperform those trained from scratch, both in terms of final performance and reduced training time despite the differences between the two domains (Morid et al., 2020; Raghu et al., 2019; Tajbakhsh et al., 2016). However, recent studies

show that, for CNNs, transfer from IMAGENET to medical tasks is not as useful as previously thought (Raghu et al., 2019; Neyshabur et al., 2020).

Supervised transfer learning requires many annotations. This can be problematic for medical tasks where annotations are costly and require specialized experts. Self-supervised approaches, on the other hand, can learn powerful representations by leveraging the intrinsic structure present in the image, rather than explicit labels (He et al., 2020; Chen et al., 2020). Although self-supervision performs best when large amounts of unlabelled data are available, it has been shown to be effective with CNNs for certain medical image analysis tasks (Azizi et al., 2021; Sowrirajan et al., 2021).

In contrast to CNNs, we know relatively little about how vision transformers perform at medical image classification. Are vanilla ViTs competitive with CNNs? Despite their success in the natural image domain, there are questions that cast doubt as to whether that success will translate to medical tasks. Evidence suggests that ViTs require very large datasets to outperform CNNs – in Dosovitskiy et al. (2020), the benefits of ViTs only became evident when Google's private 300 million image dataset, JFT-300M, was used for pretraining. Reliance on data of this scale may be a barrier to the application of transformers for medical tasks. As outlined above, CNNs rely on techniques such as transfer learning and self-supervision to perform well on medical datasets. Are these techniques as effective for transformers as they are for CNNs? Finally, the variety and type of patterns and textures in medical images differ significantly from the natural domain. Studies on CNNs indicate that the benefits from transfer diminish with the distance from the source domain (Azizpour et al., 2016; Raghu et al., 2019; Neyshabur et al., 2020) Can ViTs cope with this difference?

In this work we explore whether it is feasible for ViTs to replace CNNs for medical image classification, and if there is an advantage of doing so. Given the enormous computational cost of considering the countless variations in CNN and ViT architectures, we limit our study to prototypical CNN and ViT models over a representative set of well-known publicly available medical image benchmark datasets. Through these experiments we show that:

- ViTs pretrained on IMAGENET perform comparably to CNNs for classification tasks on medical images. While CNNs outperform ViTs when trained from scratch, ViTs receive a larger boost in performance from pretraining on IMAGENET.

- Despite their reliance on large datasets, self-supervised ViTs perform on par or better than CNNs, if IMAGENET pretraining is utilized, albeit by a small margin.

- Features of off-the-shelf IMAGENET pre-trained ViTs are versatile. $k$-NN tests show self-supervised ViT representations outperform CNNs, indicating their potential for other tasks. We also show they can serve as drop-in replacements for segmentation.

These findings, along with additional ablation studies, suggest that ViTs can be used in medical image analysis, while at the same time potentially gaining from other properties of ViTs such as built-in explainability. The source code to reproduce our work is included in the supplementary material, and we will release it at the time of publication.

## 2 RELATED WORK

**Transformers in vision problems.** Following the success of transformers (Vaswani et al., 2017a) in natural language processing (NLP), attention-based models captured the interest of the vision community and inspired numerous improvements to CNNs (Hu et al., 2018; Zhang et al., 2020). The first work to show that vanilla transformers from NLP can be applied with minimal changes to large-scale computer vision tasks and compete with standard CNNs was from Dosovitskiy et al. (2020), which introduced the term *vision transformer*, or ViT[1]. Like their NLP counterparts, the original vision transformers required an enormous corpus of training data to perform well. To overcome this problem, Touvron et al. (2021) introduced DEITs, which proposed a distillation token in addition to the CLS token that allows transformers to learn more efficiently on smaller datasets. As the number of applications of vision transformers and architectural variations have exploded (Khan et al. (2021)

---

[1]While Dosovitskiy's use of the term ViT referred to a direct adaptation of the original transformer model from NLP (Vaswani et al., 2017b), confusingly ViT has since been adopted as a term to refer to any transformer-based architecture for a vision task. We also use ViT in the latter sense, for brevity.

provides a good review), DeiT and the original ViT (Dosovitskiy et al., 2020) have become the standard benchmark architectures for vision transformers.

**Vision transformers in medical imaging.**     Given the very recent appearance of vision transformers, their application in medical image analysis has been limited. Segmentation has seen the most applications of ViTs. Chen et al. (2021a), Chang et al. (2021) and Hatamizadeh et al. (2021) independently suggested different methods for replacing CNN encoders with transformers in U-Nets (Ronneberger et al., 2015), resulting in improved performance for several medical segmentation tasks. Vanilla transformers were not used in any of these works, however. In each, modifications to the standard vision transformer from Dosovitskiy et al. (2020) are proposed, either to adapt to the U-Net framework (Chen et al., 2021a) or to add components from CNNs (Chang et al., 2021). Other transformer-based adaptations of CNN based-architectures suggested for medical segmentation tasks include Zhang et al. (2021), who combined pyramid networks with transformers, and Lopez Pinaya et al. (2021) who used transformers in variational auto encoders (VAEs) for segmentation and anomaly detection in brain MR imaging.

Only a handful of studies have tackled tasks other than segmentation, such as 3-D image registration (Chen et al., 2021b) and detection (Duong et al., 2021). The only work so far, to our knowledge, applying transformers to medical image classification is a CNN/transformer hybrid model proposed by Dai et al. (2021) applied to a small (344 images) private MRI dataset. Notably, none of these works consider pure, off-the-shelf vision transformers – all propose custom architectures combining transformer/attention modules with components from convolutional feature extractors.

**Improved initialization with transfer- and self-supervised learning.**     Medical imaging datasets are typically orders of magnitude smaller than natural image datasets due to cost, privacy concerns, and the rarity of certain diseases. A common strategy to learn good representations for smaller datasets is transfer learning. For medical imaging, it is well-known that CNNs usually benefit from transfer learning with IMAGENET (Morid et al., 2020) despite the distinct differences between the domains. However, the question of whether ViTs benefit similarly from transfer learning to medical domains has yet to be explored.

While transfer learning is one option to initialize models with good features, another more recent approach is self-supervised pre-training. Recent advances in self-supervised learning have dramatically improved performance of label-free learning. State-of-the-art methods such as DINO (Caron et al., 2021) and BYOL (Grill et al., 2020) have reached performance on par with supervised learning on IMAGENET and other standard benchmarks. While these top-performing methods have not yet been proven for medical imaging, there has been some work using earlier self-supervision schemes on medical data. Azizi et al. (2021) adopted SimCLR (Chen et al., 2020), a self-supervised contrastive learning method, to pretrain CNNs. This yielded state-of-the-art results for predictions on chest X-rays and skin lesions. Similarly, Sowrirajan et al. (2021) employed MoCo (He et al., 2020) pre-training on a target chest X-ray dataset, demonstrating again the power of this approach. While these works exhibit promising improvements thanks to self-supervision, they have all employed CNN-based encoders. It has yet to be shown how self-supervised learning combined with ViTs performs in medical imaging, and how this combination compares to its CNN counterparts.

## 3 METHODS

The main question we investigate is whether prototypical vision transformers can be used as a drop-in alternative to CNNs for medical diagnostic tasks. More concretely, we consider how each model type performs on various tasks in different domains, with different types of initialization, and across a range of model capacities. To that end, we conducted a series of experiments to compare representative ViTs and CNNs for various medical image analysis tasks under the same conditions. Hyperparameters such as weight decay and augmentations were optimized for CNNs and used for both CNNs and ViTs – with the exception of the initial learning rate, which was determined individually for each model type using a grid search.

To keep our study tractable, we selected representative CNN and ViT model types as there are too many architecture variations to consider each and every one. For CNNs, an obvious choice is the RESNET family (He et al., 2016), as it is the most common and highly cited CNN backbone, and

recent works have shown that RESNETs are competitive with more recent CNNs when modern training methods are applied (Bello et al., 2021). The choice for a representative vision transformer is less clear because the field is still developing. We considered several options including the original ViT (Dosovitskiy et al., 2020), SWIN transformers (Liu et al., 2021), CoaT (Xu et al., 2021), and focal transformers (Yang et al., 2021), but we selected the DEIT family (Touvron et al., 2021) for the following reasons: *(1)* it is one of the earliest and most established vision transformers, with the most citations aside from the original ViT, *(2)* it is similar in spirit to a pure transformer, *(3)* it was the first to show that vision transformers can compete on mid-sized datasets with short training times, *(4)* it retains the interpretability properties of the original transformer.

As mentioned above, CNNs rely on initialization strategies to improve performance for smaller datasets. Accordingly, we consider three common initialization strategies: randomly initialized weights, transfer learning from IMAGENET, and self-supervised pretraining on the target dataset. Using these models and initialization strategies, we consider the following datasets and tasks:

**Medical image classification.** Five standard medical image classification datasets were chosen to be representative of a diverse set of target domains. These cover different imaging modalities, color distributions, dataset sizes, and tasks, with expert labels.

- **APTOS 2019** – In this dataset, the task is classification of diabetic retinopathy images into 5 categories of disease severity (Kaggle, 2019). APTOS 2019 contains 3,662 high-resolution retinal images.
- **CBIS-DDSM** – A mammography dataset containing 10,239 images. The task is to detect the presence of masses in the mammograms (Sawyer-Lee et al., 2016; Lee et al., 2017; Clark et al., 2013).
- **ISIC 2019** – Here, the task is to classify 25,333 dermoscopic images among nine different diagnostic categories of skin lesions (Tschandl et al., 2018; Codella et al., 2018; Combalia et al., 2019).
- **CheXpert** – This dataset contains 224,316 chest X-rays with labels over 14 categories of diagnostic observations (Irvin et al., 2019).
- **PatchCamelyon** – Sourced from the Camelyon16 segmentation challenge (Bejnordi et al., 2017), this dataset contains 327,680 patches of H&E stained WSIs of sentinel lymph node sections. The task is to classify each patch as cancerous or normal (Veeling et al., 2018).

**Medical image segmentation.** Although there has been some recent work applying vision transformers to medical image segmentation, all works thus far have proposed hybrid architectures. Here, we evaluate the ability of vanilla ViTs to directly replace the encoder of a standard segmentation framework designed for CNNs. In particular, we use DEEPLAB3 (Chen et al., 2017) featuring a RESNET50 encoder. We merely replace the encoder of the segmentation model with DEIT-S and feed the outputs of the CLS token to the decoder of DEEPLAB3. As DEEPLAB3 was designed and optimized for RESNETs, CNNs have a clear advantage in this setup. Nevertheless, these experiments can shed some light into the quality and versatility of ViT representations for segmentation. Note that, unlike the classification experiments, all models were initialized with IMAGENET pretrained weights. We consider the following medical image segmentation datasets:

- **ISIC 2018** – This dataset consists of 2,694 dermoscopic images with their corresponding pixel-wise binary annotations. The task is to segment skin lesions (Codella et al., 2019; Tschandl et al., 2018).
- **CSAW-S** – This dataset contains 338 images of mammographic screenings. The task is pixel-wise segmentation of tumor masses (Matsoukas et al., 2020).

**Experimental setup.** We selected the RESNET family (He et al., 2016) as representative CNNs, and DEIT[2] (Touvron et al., 2021) as representative ViTs. Specifically, we test RESNET-18, RESNET-50, and RESNET-152 against DEIT-T, DEIT-S, and DEIT-B. These models were chosen because they are easily comparable in the number of parameters, memory requirements, and compute. They are also some of simplest, most commonly used, and versatile architectures of their respective type. Note that ViTs have an additional parameter, patch size, that directly influences the memory and computational requirement. We use the default $16 \times 16$ patches to keep the number of parameters comparable to the CNN models.

---

[2]Note that we refer to the DEIT architecture pretrained *without* the distillation token (Touvron et al., 2021).

| Initialization | Model | APTOS2019, $\kappa$ ↑ $n = 3{,}662$ | DDSM, ROC-AUC ↑ $n = 10{,}239$ | ISIC2019, Recall ↑ $n = 25{,}333$ | CheXpert, ROC-AUC ↑ $n = 224{,}316$ | Camelyon, ROC-AUC ↑ $n = 327{,}680$ |
|---|---|---|---|---|---|---|
| Random | ResNet50 | $0.849 \pm 0.022$ | $0.916 \pm 0.005$ | $0.660 \pm 0.016$ | $0.796 \pm 0.000$ | $0.943 \pm 0.008$ |
|  | DeiT-S | $0.687 \pm 0.017$ | $0.906 \pm 0.005$ | $0.579 \pm 0.013$ | $0.762 \pm 0.002$ | $0.921 \pm 0.002$ |
| ImageNet (supervised) | ResNet50 | $0.893 \pm 0.004$ | $0.954 \pm 0.005$ | $0.810 \pm 0.008$ | $0.801 \pm 0.001$ | $0.960 \pm 0.004$ |
|  | DeiT-S | $0.896 \pm 0.005$ | $0.949 \pm 0.006$ | $0.844 \pm 0.021$ | $0.794 \pm 0.000$ | $0.964 \pm 0.008$ |
| ImageNet (supervised) + self-supervised with DINO | ResNet50 | $0.894 \pm 0.008$ | $0.955 \pm 0.002$ | $0.833 \pm 0.007$ | $0.801 \pm 0.000$ | $0.962 \pm 0.004$ |
|  | DeiT-S | $0.896 \pm 0.010$ | $0.956 \pm 0.002$ | $0.853 \pm 0.009$ | $0.801 \pm 0.001$ | $0.976 \pm 0.002$ |

Table 1: *Comparison of vanilla CNNs vs. ViTs with different initialization strategies on medical image classification tasks.*

Using these models, we compare three commonly used initialization strategies:

1. Randomly initialized weights (Kaiming initialization from He et al. (2015)),

2. Transfer learning using supervised IMAGENET pretrained weights,

3. Self-supervised pretraining on the *target dataset*, following IMAGENET initialization.

After initialization using the strategies above, the models are fine-tuned on the target data using the procedure described below. Early experiments indicated that initialization using self-supervision on the target dataset consistently outperforms self-supervision on IMAGENET. We also determined that DINO performs better than other self-supervision strategies such as BYOL (Grill et al., 2020).

**Training procedure.** For supervised training, we use the ADAM optimizer (Kingma & Ba, 2014)[3]. We performed independent grid searches to find suitable learning rates, and found that $10^{-4}$ works best for both pretrained CNNs and ViTs, while $10^{-3}$ is best for random initialization. We used these as base learning rates for the optimizer along with default 1,000 warm-up iterations. When the validation metrics saturated, the learning rate was dropped by a factor of 10 until it reached its final value of $10^{-6}$. For classification tasks, images were resized to $256 \times 256$ with the following augmentations applied: normalization; color jitter which consists of brightness, contrast, saturation, hue; horizontal flip; vertical flip; and random resized crops. For segmentation, all images were resized to $512 \times 512$ and we use the same augmentation methods as for classification except for CSAW-S, where elastic transforms are also employed. For each run, we select the checkpoint with highest validation performance. For self-supervision, pretraining starts with IMAGENET initialization, then applies DINO (Caron et al., 2021) on the target data following the default settings – except for three small changes determined to work better on medical data: *(1)* the base learning rate was set to $10^{-4}$, *(2)* the initial weight decay is set at $10^{-5}$ and increased to $10^{-4}$ using a cosine schedule, and *(3)* we used an EMA of 0.99. The same settings were used for both CNNs and ViTs; both were pre-trained for 300 epochs using a batch size of 256, followed by fine-tuning as described above.

The classification datasets were divided into train/test/validation splits (80/10/10), with the exception of APTOS2019, which was divided 70/15/15 due to its small size. For the segmentation datasets, we split the training set into train/validation with a 90/10 ratio. For ISIC2018 and CSAW-S, we used the provided validation and test sets for model evaluation.

**Evaluation.** Unless otherwise specified, each experimental condition was repeated five times. We report the median and standard deviation of the appropriate metric for each dataset: Cohen Kappa, Recall, ROC-AUC, and IoU. For classification, we measure performance of the:

- initialized model representations, through a $k$-NN evaluation protocol (Caron et al., 2021)
- final model after *fine-tuning* on the target dataset.

The $k$-NN evaluation protocol is a technique to measure the usefulness of learned representations for some target task; it is typically applied before fine-tuning. For CNNs, it works by freezing the pretrained encoder, passing test examples through the network, and performing a $k$-NN classification using the cosine similarity between embeddings of the test images and the $k$ nearest training images ($k = 200$ in our case). For ViTs, the principle is the same except the output of the CLS token is used instead of the CNN's penultimate layer.

---

[3] DEIT trained with random initialization used AdamW (Loshchilov & Hutter, 2017) instead of Adam.

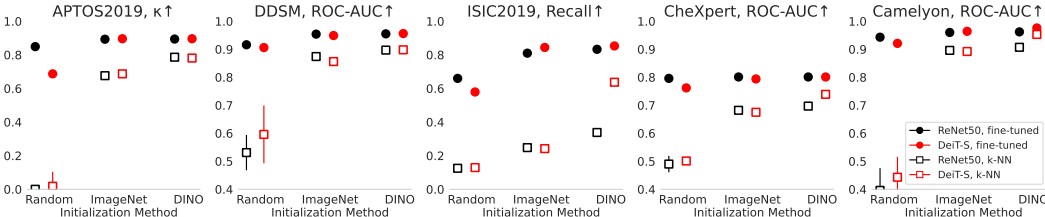

Figure 1: Performance comparison of RESNET50 and DEIT-S, two commonly used CNN-based and ViT-based architectures. The comparison covers several standard medical image classification datasets and different types of initialization including random init, IMAGENET pretraining, and self-supervision using DINO (Caron et al., 2021). Performance is measured after fine-tuning on the dataset, as well as using $k$-NN evaluation without fine-tuning. We report the median over 5 repetitions, error bars represent standard deviation. Numeric values appear in Table 3 (Appendix A)

## 4  EXPERIMENTAL RESULTS

**Are randomly initialized vision transformers useful?**   Our first experiment compares the performance of ViTs against CNNs when initialized with random weights (He et al., 2015). The results in Table 1 and Figure 1 indicate that in this setting, CNNs outperform ViTs across the board. This is in line with previous observations in the natural image domain, where ViTs trained on smaller datasets are outperformed by similarly-sized CNNs, a trend that was attributed to the vision transformer's lack of inductive bias (Dosovitskiy et al., 2020). Note that the performance gap appears to shrink as the number of training examples $n$ increases. However, since most medical imaging datasets are of modest size, the usefulness of randomly initialized ViTs appears to be limited.

**Does pretraining transformers on IMAGENET work in the medical image domain?**   To deal with the smaller size of medical imaging datasets, CNNs are often initialized with IMAGENET pretrained weights. This typically results in better performance than random initialization (Morid et al., 2020), while recent works have questioned the usefulness of this approach for CNNs (Raghu et al., 2019; Neyshabur et al., 2020). We investigate if ViTs benefit from IMAGENET pre-training in the medical domain and to which extent. To test this, we initialize all models with supervised IMA-GENET-pretrained weights and then fine-tune on the target data using the procedure described in Section 3. The results in Table 1 and Figure 1 show that both CNNs and ViTs benefit from IMA-GENET initialization. However, ViTs appear to benefit more from transfer learning, as they make up for the gap observed using random initialization, performing on par with their CNN counterparts.

**Do transformers benefit from self-supervision in the medical image domain?**   Recent self-supervised learning schemes such as DINO and BYOL achieve performance near that of supervised learning. When used for pretraining in combination with supervised fine-tuning, they can achieve a new state-of-the-art (Caron et al., 2021; Grill et al., 2020). While this phenomenon has been demonstrated for CNNs and ViTs on big data in the natural image domain, it is not clear whether self-supervised pretraining of ViTs helps for medical imaging tasks. To test if ViTs benefit from self-supervision, we adopt the learning scheme of DINO, which can be readily applied to both CNNs and ViTs (Caron et al., 2021). The results reported in Table 1 and Figure 1 show that both ViTs and CNNs perform better with self-supervised pretraining. ViTs appear to perform on par, or better than CNNs in this setting, albeit by a small margin.

**Are pretrained ViT features useful for medical imaging, even without fine-tuning?**   When data is particularly scarce, features from pre-trained networks can sometimes prove useful, even without fine-tuning on the target domain. The low dimensional embeddings from a good feature extractor can be used for tasks such as clustering or few-shot classification. We investigate whether different types of pretraining in ViTs yields useful representations. We measure this by applying the $k$-NN evaluation protocol described in Section 3. The embeddings from the penultimate layer of the CNN and the CLS token of the ViT are used to perform $k$-NN classification of test images, assigning labels based on the most similar examples from the training set. We test whether out-of-domain IMAGENET pretraining or in-domain self-supervised DINO pretraining (Caron et al., 2021) yields useful representations when no supervised fine-tuning has been done. The results

| Model | ISIC2018, IoU ↑ | CSAW-S, IoU ↑ |
|---|---|---|
| DEEPLAB3-RESNET50 | $0.802 \pm 0.012$ | $0.320 \pm 0.008$ |
| DEEPLAB3-DEIT-S | $0.845 \pm 0.014$ | $0.322 \pm 0.028$ |

Table 2: *Medical image segmentation with* DEEPLAB3 *comparing CNN vs. ViT encoders.*

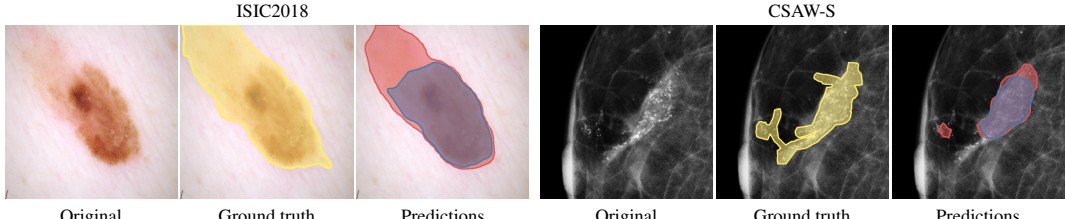

Figure 2: Medical image segmentation results comparing DEEPLAB3-RESNET50 (blue), DEEPLAB3-DEIT-S (red). Ground truth mask appears in yellow. Note that the ViT segmentations tend to do a better job of segmenting distant regions.

appear in Figure 1 and Table 3 of the Appendix A, where we compare embeddings for DEIT-S and RESNET50. Pretraining with IMAGENET yields surprisingly useful features for both CNNs and ViTs, considering that the embeddings were learned on out-of-domain data. We observe even more impressive results with in-domain self-supervised DINO pretraining. ViTs appear to benefit more from self-supervision than CNNs – in fact, ViTs trained without any labels using DINO perform similar or better than the same model trained from scratch with full supervision. The results indicate that representations from pretrained vision transformers, both in-domain and out-of domain, can serve as a solid starting point for transfer learning or $k$-shot methods.

**Do ViTs learn meaningful representations for other tasks like segmentation?** Segmentation is an important medical image analysis task. While previous works have shown that custom attention-based segmentation models can outperform CNNs (see Section 2), here we ask: *can we achieve comparable performance to CNNs by merely replacing the encoder of a CNN-based segmentation model with a vision transformer?* This would demonstrate versatility of ViT representations for other tasks. We consider DEEPLAB3 (Chen et al., 2017), a mainstream segmentation model with a RESNET50 encoder. We simply replace the RESNET50 with DEIT-S, and train the model as prescribed in Section 3. Despite the model being designed for RESNET50 and DEIT-S having fewer parameters connected to the DEEPLAB3 head, we observe that ViTs perform on par or better than CNNs in our segmentation tasks. The results in Table 2 and Figure 2 clearly indicate that vision transformers are able to produce high quality embeddings for segmentation, even in this disadvantaged setting.

**How does capacity of ViT models affect medical image classification?** To understand the practical implications of ViT model size for medical image classification, we conducted an ablation study comparing different model capacities of CNNs and ViTs. All models were initialized with IMAGENET pretrained weights, and we followed the training settings outlined in Section 3. The results are presented in Figure 3 and Table 4 in Appendix B. Both CNNs and ViTs seem to benefit similarly from increased model capacity, within the margin of error. For most of the datasets, switching from a small ViT to a bigger model results in minor improvements, with the exception of ISIC 2019 where model size appears to be an important factor for optimal performance and APTOS2019 where DeiT-T performs better than its larger variants. Perhaps the tiny size of APTOS2019 results in diminishing returns for large models.

**Other ablation studies.** We also considered the effect of token size on ViT performance. Table 5 in Appendix C shows that reducing patch size results in a slight boost in performance, but at the cost of increasing the memory footprint. Finally, we investigate how ViTs attend to different regions of medical images. In Figure 5 of Appendix D we compute the *attention distance*, the average distance in the image across which information is integrated in ViTs. We find that, in early layers, some heads attend to local regions and some to the entire image. Deeper into the network, attention becomes more global – though more quickly for some datasets than for others.

Figure 3: Impact of model capacity on performance for the RESNET and DEIT families on standard medical image classification datasets. Both model types seem to perform better with increasing capacity, roughly scaling similarly. Numeric results appear in Table 4 of Appendix B.

## 5 DISCUSSION

Our investigation of prototypical CNNs and ViTs *indicate that CNNs can readily be replaced with vision transformers in medical imaging tasks without sacrificing performance.* The caveat being that employing some form of transfer learning is necessary. Our experiments corroborate previous findings in the natural image domain and provide new insights, which we discuss below.

**Discussion and implications of findings.** Unsurprisingly, we found that CNNs outperform ViTs when trained from scratch on medical image datasets, which corroborates previous findings in the natural image domain (Dosovitskiy et al., 2020). This trend appears consistently and fits with the "insufficient data to overcome the lack of inductive bias" argument. Thus, the usefulness of randomly initialized ViTs appears to be limited in the medical imaging domain.

When initialized with supervised IMAGENET pretrained weights, the gap between CNN and ViT performance disappears on medical tasks. The benefits of supervised IMAGENET pretraining of CNNs is well-known, but it was unexpected that ViTs would benefit so strongly. To the best of our knowledge, we are the first to confirm that supervised IMAGENET pretraining is so effective for ViTs in the medical domain. This suggests that further improvements could be gained via transfer learning from other domains more closely related to the task, as is the case for CNNs (Azizpour et al., 2016). The benefits of IMAGENET pretraining was not only observed for the fine-tuned classification tasks, but also self-supervised learning, $k$-nn classification and segmentation.

The best overall performance on medical images is obtained using ViTs with in-domain self-supervision, where small improvements over CNNs and other initialization methods were recorded. Our $k$-NN evaluation showed that ViT features learned this way are even strong enough to outperform supervised learning with random initialization. Interestingly, we did not observe as strong of an advantage for self-supervised ViTs over IMAGENET pretraining as was previously reported in the natural image domain, *e.g.* in Caron et al. (2021). We suspect this is due to the limited size of medical datasets, suggesting a tantalizing opportunity to apply self-supervision on larger and easier to obtain unlabeled medical image datasets, where greater benefits may appear.

Our other ablation studies showed that ViT performance scales with capacity in a similar manner to CNNs; that ViT performance can be improved by reducing the patch size; and that ViTs attend to information across the whole image even in the earliest layers. Although individually unsurprising, it is worthwhile to confirm these findings for medical images. It should be noted however, that the memory demands of DeiTs increase quadratically with the image and patch size which might limit their application in large medical images. However, newer ViT architectures (e.g. Liu et al. (2021) and Yang et al. (2021)) mitigate this issue – while demonstrating increased predictive performance.

**Interesting properties of ViTs.** Our results indicate that switching from CNNs to ViTs can be done without compromising performance, but *are there any other advantages to switching to ViTs?* There are a number of important differences between CNNs and ViTs. We briefly discuss some of these differences and why they may be interesting for medical image analysis.

*Lack of inductive bias* – ViTs do away with convolutional layers. Our experiments indicate their implicit locality bias is only necessary when training from scratch on medical datasets. For larger datasets, evidence suggests removing this bias improves performance (Dosovitskiy et al., 2020).

*Global + local features* – ViTs, unlike CNNs, can combine information from distant and local regions of the image, even in early layers (see Appendix D). This information can be propagated

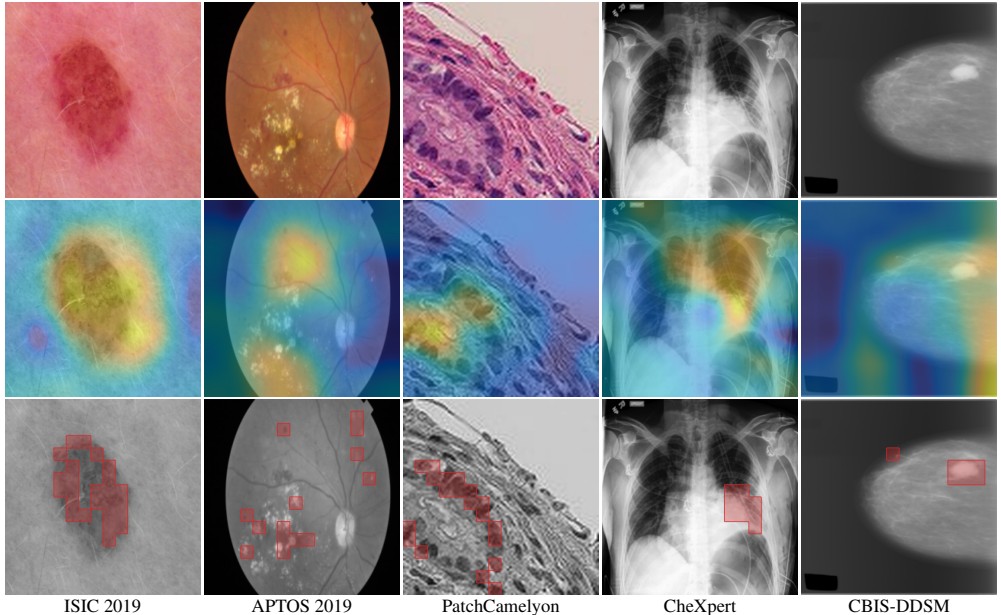

ISIC 2019          APTOS 2019          PatchCamelyon          CheXpert          CBIS-DDSM

Figure 4: Comparing saliency for RESNET50 ($2^{nd}$ row) and DEIT-S ($3^{rd}$ row) on medical classification. Each column contains the original, a Grad-CAM visualization visualisation for ResNet50 (Selvaraju et al., 2017) and the top-$50\%$ attention map of the CLS token of DEIT-S.

efficiently to later layers due to better utilization of residual connections that are unpolluted by pooling layers. This may prove beneficial for medical modalities which rely on local features. It may be necessary to use transfer learning to benefit from local features, as they require huge amounts of data to learn (Raghu et al., 2021).

*Interpretability* – transformers' self-attention mechanism provides, for free, new insight into how the model makes decisions. CNNs do not naturally lend themselves well to visualizing saliency. Popular CNN explainability methods such as class activation maps (Zhou et al., 2016) and Grad-CAM (Selvaraju et al., 2017) provide coarse visualizations because of pooled layers. Transformer tokens give a finer picture of attention, and the self-attention maps explicitly model interactions between every region in the image. In Figure 4, we show examples from each dataset along with Grad-CAM visualizations of RESNET-50 and the top-50% self-attention of $16 \times 16$ DEIT-S CLS token heads. The ViTs provide a clear, localized picture of attention, *e.g.* attention at the boundary of the skin lesion in ISIC, on hemorrhages and exudates in APTOS, the membrane of the lymphatic nodule in PatchCamelyon, the opacity in the Chest X-ray, and a dense region in CBIS-DDSM.

## 6 CONCLUSION

To answer the question posed in the title: *vanilla transformers can reliably replace CNNs on medical image classification with little effort.* More precisely, ViTs reach the same level of performance as CNNs in an array of medical classification and segmentation tasks, but they require transfer learning to do so. But since IMAGENET pretraining is the standard approach for CNNs, this does not incur any additional cost in practice. The best overall performance on medical imaging tasks is achieved using in-domain self-supervised pretraining, where ViTs show a small advantage over CNNs. As the data size grows, this advantage is expected to grow as well, as observed by Caron et al. (2021) and our own experiments. Additionally, ViTs have a number of properties that make them attractive: they scale similarly to CNNs (or better), their lack of inductive bias, global attention and skip connections may improve performance, and their self-attention mechanism provides a clearer picture of saliency. From a practitioner's point of view, these benefits are compelling enough to explore the use of ViTs in the medical domain. Finally, modern CNNs have been studied extensively for 15 years, while the first pure vision transformer appeared one year ago – the potential for ViTs to improve is considerable.

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

APPENDIX

The Appendix contains several details and additional experimental results that did not fit within the page limit, organized by the order in which they are referenced in the main article.

## A $k$-NN EVALUATION OF DEIT-S AND RESNET50

The $k$-NN evaluation protocol is a technique to measure the usefulness of learned representations for some target task; it is typically applied before fine-tuning. For CNNs, it works by freezing the pretrained encoder, passing test examples through the network, and performing a $k$-NN classification using the cosine similarity between embeddings of the test images and the $k$ nearest training images ($k = 200$ in our case). For ViTs, the principle is the same except the output of the CLS token is used instead of the CNN's penultimate layer.

We applied the $k$-NN evaluation protocol to test whether out-of-domain IMAGENET pretraining or in-domain self-supervised DINO pretraining yields useful representations when no supervised fine-tuning has been done. The results appear in Table 3. Pretraining with IMAGENET yields surprisingly useful features for both CNNs and ViTs, considering that the embeddings were learned on out-of-domain data. We observe even more impressive results with in-domain self-supervised DINO pretraining. ViTs appear to benefit more from self-supervision than CNNs – in fact, ViTs trained with DINO perform similar or better than the same model trained from scratch with full supervision! The results indicate that representations from pretrained vision transformers, both in-domain and out-of domain, can serve as a solid starting point for transfer learning or $k$-shot methods.

| Initialization | Model | APTOS2019, $\kappa$ ↑ | DDSM, ROC-AUC ↑ | ISIC2019, Recall ↑ | CheXpert, ROC-AUC ↑ | Camelyon, ROC-AUC ↑ |
|---|---|---|---|---|---|---|
| Random | ResNet50 | $0.000 \pm 0.000$ | $0.531 \pm 0.063$ | $0.125 \pm 0.000$ | $0.490 \pm 0.029$ | $0.396 \pm 0.079$ |
| | DeiT-S | $0.018 \pm 0.086$ | $0.596 \pm 0.103$ | $0.129 \pm 0.026$ | $0.501 \pm 0.013$ | $0.443 \pm 0.073$ |
| ImageNet (supervised) | ResNet50 | $0.676 \pm 0.000$ | $0.874 \pm 0.000$ | $0.248 \pm 0.000$ | $0.682 \pm 0.000$ | $0.896 \pm 0.000$ |
| | DeiT-S | $0.687 \pm 0.000$ | $0.856 \pm 0.000$ | $0.242 \pm 0.000$ | $0.675 \pm 0.000$ | $0.892 \pm 0.000$ |
| ImageNet (supervised) + | ResNet50 | $0.786 \pm 0.000$ | $0.897 \pm 0.000$ | $0.338 \pm 0.000$ | $0.697 \pm 0.000$ | $0.907 \pm 0.000$ |
| self-supervised with DINO | DeiT-S | $0.781 \pm 0.000$ | $0.898 \pm 0.000$ | $0.637 \pm 0.000$ | $0.739 \pm 0.000$ | $0.953 \pm 0.000$ |

Table 3: The $k$-NN evaluation of CNNs and ViTs with different initialization methods including random initialization, IMAGENET pretraining, and self-supervision using DINO (Caron et al., 2021) on the target medical dataset. For each task, we use the metrics that are commonly used in the literature, and for random initialization we report the median ($\pm$ standard deviation) over 5 repetitions. The numbers in this table correspond to Figure 1.

## B DEIT AND RESNET WITH DIFFERENT CAPACITIES

To understand the practical implications of ViT model size for medical image classification, we compare different model capacities of CNNs and ViTs.

Specifically, we test RESNET-18, RESNET-50, and RESNET-152 against DEIT-T, DEIT-S, and DEIT-B. These models were chosen because they are easily comparable in the number of parameters, memory requirements, and compute. They are also some of simplest, most commonly used, and versatile architectures of their respective type. All models were initialized with IMAGENET pretrained weights, and trained following the settings outlined in Section 3. The results appear in Table 4.

Both CNNs and ViTs seem to benefit similarly from increased model capacity, within the margin of error. For most of the datasets, switching from a small ViT to a bigger model results in minor improvements, with the exception of ISIC 2019 where model size appears to be an important factor for optimal performance.

| Model | APTOS2019, $\kappa$ ↑ | DDSM, ROC-AUC ↑ | ISIC2019, Recall ↑ | CheXpert, ROC-AUC ↑ | Camelyon, ROC-AUC ↑ |
|---|---|---|---|---|---|
| ResNet18 | 0.893 ± 0.003 | 0.950 ± 0.003 | 0.785 ± 0.015 | 0.793 ± 0.001 | 0.959 ± 0.012 |
| ResNet50 | 0.893 ± 0.004 | 0.954 ± 0.005 | 0.810 ± 0.008 | 0.801 ± 0.001 | 0.960 ± 0.004 |
| ResNet152 | 0.900 ± 0.004 | 0.960 ± 0.003 | 0.798 ± 0.012 | 0.801 ± 0.001 | 0.965 ± 0.003 |
| DeiT-T | 0.901 ± 0.005 | 0.946 ± 0.006 | 0.810 ± 0.013 | 0.789 ± 0.001 | 0.961 ± 0.003 |
| DeiT-S | 0.896 ± 0.005 | 0.949 ± 0.006 | 0.844 ± 0.021 | 0.794 ± 0.000 | 0.964 ± 0.008 |
| DeiT-B | 0.897 ± 0.004 | 0.953 ± 0.004 | 0.840 ± 0.013 | 0.795 ± 0.001 | 0.969 ± 0.002 |

Table 4: Effect of model capacity on the performance after fine-tuning, with three different RESNET variants (RESNET18, RESNET50 and RESNET152) and DEIT variants (DEIT-T, DEIT-S and DEIT-B ) using IMAGENET pretraining. For each task, we report the median (± standard deviation) over 5 repetitions using the metrics that are commonly used in the literature. We can see that both CNNs and ViTs mostly benefit from increased model capacity. Further, model types with similar capacity perform approximately on par in most of the datasets, indicating that ViTs scale similarly to CNNs. The numbers in this table correspond to Figure 3.

## C EFFECT OF PATCH SIZE

ViTs have an additional parameter, patch size, that directly influences the memory and computational requirement. We compare the default $16 \times 16$ patches, used in this work, with $8 \times 8$ patches, as it has been noted in previous works that reducing the patch size leads to increased performance (Caron et al., 2021).

The results appear in Table 5. We observe that smaller patch sizes result in marginally better performance, though to a lesser degree than has been observed in the natural image domain.

| Patch size | APTOS2019, $\kappa$ ↑ | DDSM, ROC-AUC ↑ | ISIC2019, Recall ↑ | CheXpert, ROC-AUC ↑ | Camelyon, ROC-AUC ↑ |
|---|---|---|---|---|---|
| $16 \times 16$ | 0.896 ± 0.005 | 0.949 ± 0.006 | 0.844 ± 0.021 | 0.794 ± 0.000 | 0.964 ± 0.008 |
| $8 \times 8$ | 0.898 ± 0.006 | 0.947 ± 0.007 | 0.847 ± 0.015 | 0.800 ± 0.001 | 0.964 ± 0.002 |

Table 5: Comparison of performance of DEIT-S using $8 \times 8$ and $16 \times 16$ pixel input patches on standard medical image classification datasets. Performance is measured after fine-tuning on the dataset. For each task we report the median (± standard deviation) over 5 repetitions using the metrics that are commonly used in the literature. Although, the model using $8 \times 8$ patch sizes median performance is slightly better in the majority of datasets, we see no significant improvements, aside from CheXpert.

## D MEAN ATTENDED DISTANCE OF VITS

We compute the *attention distance*, the average distance in the image across which information is integrated in ViTs. As reported previously (e.g. Dosovitskiy et al. (2020)), in early layers, some heads attend to local regions and some to the entire image. Deeper into the network, attention becomes more global – though more quickly for some datasets than for others.

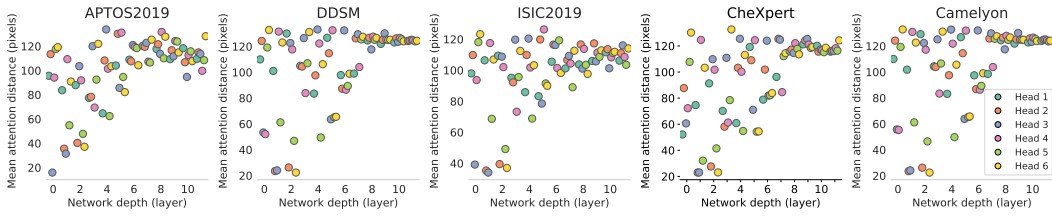

Figure 5: Mean attention distance with respect to the attention head and the network depth. Each point is calculated as the average over 512 test samples as the mean of the element-wise multiplication of each query token's attention and its distance from the other tokens

