# OpenReview forum: "Should we Replace CNNs with Transformers for Medical Images?"
_ICLR.cc/2022/Conference — ICLR 2022 Submitted_

### Official Review · Reviewer_LKZi · 2021-10-28

**Correctness:** 4
**Technical Novelty And Significance:** 1
**Empirical Novelty And Significance:** 2
**Recommendation:** 3
**Confidence:** 4

**Main Review:**

# Strengths:
   - This paper performs tests in multiple datasets for two common types of tasks in the medical imaging domain: classification and segmentation.  It also evaluates CNNs and ViTs models that are very representative like ResNets and DEIT.
- The paper is also well written and easy to follow.

# Weaknesses:
- I feel some simple analyses are missing: For instance, how long take to train each model? What are their inference time? Which one uses less GPU memory during training and inference? How do these models behave as we increase the size of the training dataset?
- There are no fine-grained qualitative experiments: What kind of mistakes are more common in each model. For instance, CNNs in segmentation are known to not segment well very small structures, does it also happen with ViTs?
- Use of more challenging datasets for segmentation: brain segmentation would be a good benchmark since it has more challenging structures and a lot more classes than the ones used.


**Summary Of The Paper:**

This paper describes experiments comparing convolutional neural networks (CNNs) and visual transformers (ViTs) models in classification and segmentation problems in medical imaging. They also evaluate how the initialization of these models is essential to reach good performance in these problems which suffer from a lack of labeled data. In these experiments, they conclude that, ViTs can reach the same level of performance as CNNs, but they require transfer learning to do so, which can be easily accomplished with IMAGENET pertaining (like CNN's usually does) or self-supervised learning.

**Summary Of The Review:**

I think this paper is not ready for publication in a top venue like ICLR. This paper does not present any technical novelty since it is an empirical paper. I agree that empirical papers can contribute a lot to the community as well, but it requires deep analysis of the results. The experiments are very coarse only comparing the average performance of two models in classification and segmentation of medical imaging. In my opinion, the findings of these experiments are somewhat expected and even the difference between the two models are too small to conclude something. Due to these reasons, I suggest rejecting this paper.

---

> ### Author Response · Authors · 2021-11-20
> **Response to Reviewer LKZi**
>
> We thank the reviewer for their feedback. Unfortunately, it seems that we have not effectively communicated our findings to this reviewer. Please see our comment to all reviewers above clarifying the contributions of our work and additional experiments. We address your comments below.
>
> 1. ***“I feel some simple analyses are missing: For instance, how long take to train each model? What are their inference time? Which one uses less GPU memory during training and inference? How do these models behave as we increase the size of the training dataset?”***
> We selected models that have similar computational and memory demands for a fair comparison. However, DeiTs do not scale well with the image size. This is one of the reasons that led to the creation of other ViT models, like SWIN, which scales linearly. We added this to the discussion. We show how that, regardless the dataset size, ViTs can perform comparably with the CNNs if transfer learning and SSL is utilized.
>
>
> 2. ***There are no fine-grained qualitative experiments: What kind of mistakes are more common in each model. For instance, CNNs in segmentation are known to not segment well very small structures, does it also happen with ViTs?”***
> While we agree that investigation into modes of failure is interesting, such an investigation would necessarily be specific to each dataset and it would be difficult to draw general conclusions. W.r.t. the suggestion investigating segmentation of small structures: DeiT, the version of the ViTs that we use, struggles with very small objects due to the 16x16 patch size. SWIN and Focal transformers solve this problem and they consistently outperform CNNs across the board (see the work of (Liu et al., 2021) and (Yang et al., 2021) for more details).
>
>
> 3. ***“Use of more challenging datasets for segmentation: brain segmentation would be a good benchmark since it has more challenging structures and a lot more classes than the ones used.”***
> In Section 3 we justify a selection of 7 representative datasets and tasks across a range of modalities to show an interesting phenomenon. Yes, including brain segmentation would have been interesting as well. It is not uncommon for reviewers to ask for experiments with more datasets, but we believe we have covered a sufficiently broad set of modalities to demonstrate our findings.
>
>
> 4. ***“In my opinion, the findings of these experiments are somewhat expected and even the difference between the two models are too small to conclude something.”***
> We respectfully disagree that the findings are to be expected. While your intuition may have predicted our findings (as did ours), this conclusion was not obvious according to the current literature. The current understanding is that pure ViTs only perform well when huge amounts of data are available (Dosovitskiyet al. (2020)) and transfer learning from ImageNet to the medical domain and other distant domains  is not as useful as previously thought (Raghu et al., 2019, Neyshabur et al., 2021). Considering the size of medical datasets and their distance to the natural domain, the literature suggests that ViTs struggle to perform well in the medical domain. Self-supervised learning with ViTs has only been tested with ImageNet-size datasets (Caron et al., 2021), so this angle remains an open question. From this perspective, our findings that ViTs are able to perform comparably with CNNs and that self-supervised methods can also be deployed in the medical domain if transfer learning from ImageNet is utilized, challenge the thinking in these prior works.

---

### Official Review · Reviewer_taLm · 2021-11-01

**Correctness:** 3
**Technical Novelty And Significance:** 1
**Empirical Novelty And Significance:** 3
**Recommendation:** 6
**Confidence:** 5

**Main Review:**

Strengths:

-> It asks and answers some important questions in the current research scenario like are vanilla ViTs really competitive with CNNs?; Is transfer learning and self supervision as effective for transformers as they are for CNNs? Etc.
-> This paper finds out that self-supervised pre-training on domain-specific medical images followed by supervised fine-tuning of ViTs gives the best performance. This could be useful in future works, especially in applications like clinical deployment.

Concerns:

-> The authors choose one representation for both CNN and transformer model to study the difference. It is definitely true that not all the architecture variations could be considered for this study. However, I feel that the authors should have chosen U-Net instead of Resnet family as the representative CNN architecture for segmentation. While Resnet is a strong baseline for classification, almost all the medical segmentation methods are built with U-Net as the backbone (Litjens et al. “A survey on deep learning in medical image analysis” ).  While the authors give 4 reasons on why they chose DeiT as their transformer, there is no reason why resnet was chosen over U-Net.

-> There seems to be a significant difference in performance between DeiT and resnet on the ISIC classification task when compared to the other classification tasks (from Table 1). It would be interesting if the authors could shed some light on this observation.

-> It is shown in Isensee et al. (“nnU-Net: a self-configuring method for deep learning-based biomedical image segmentation”) that most of the performance improvement comes for medical imaging in choosing the perfect pre-processing, post-processing, training and optimization strategy of the network. I think it is important to see if nnU-Net outperforms DeiT in similar settings. If not, the authors should give a good reason why that comparison is not necessary.

-> For the segmentation experiments, the authors consider replacing the encoder part of DeepLab with DeiT which makes the case that transformers make strong encoders for segmentation. A more thorough analysis would include experimenting with a transformer based decoder. Is there any specific reason why the authors left that out, if yes, it would be nice to see that information in the paper.

-> In the discussion section “Interesting Properties of ViTs”, it would be nice to see some emphasis on proving it from the medical imaging perspective for the “Global+Local features” section.

Minor Suggestions:

-> If the page limit isn’t exceeded, It might be better to include a separate table on the findings and conclusions of this paper. For example, there could be a table with CNN and transformer as the columns and the rows corresponding to different settings (like random init, self-supervised etc.). That table will help improve the readability of the paper.

-> I think a clear difference between the findings of this paper and findings of Azizi et al. (which was on self-supervision and transfer learning for CNNs would be good to tabulate.

Post rebuttal:

I thank the reviewers for their efforts put in the reply. I think the reviewers have addressed my main concerns well. I am also satisfied with most of their responses to other reviewers as well. I am increasing my score.


**Summary Of The Paper:**

This paper primarily analyses if we should replace CNNs with transformers for medical imaging. Experiments are conducted on a number of medical image benchmark datasets with both CNNs and transformers for both classification and segmentation. It is concluded that ViTs outperform CNNs when pretrained using self-supervision.

**Summary Of The Review:**

I think the paper has some issues regarding representation for both CNN and transformer models to study the difference. As the main contribution of the paper is empirical, this becomes a main issue. The empirical findings of paper like paper self-supervised pre-training on domain-specific medical images followed by supervised fine-tuning of ViTs gives the best performance could be useful to the research community.

---

> ### Author Response · Authors · 2021-11-20
> **Response to Reviewer taLm (Part 2)**
>
>
>
> 8. ***“If the page limit isn’t exceeded, It might be better to include a separate table on the findings and conclusions of this paper ...”***
> Given the page limit it is impossible to add this in the main paper.
>
>
> 9. ***“I think a clear difference between the findings of this paper and findings of Azizi et al. (which was on self-supervision and transfer learning for CNNs would be good to tabulate.”***
> None of our results contradict those of Azizi et al. In fact, our best performing models follow a variation of their approach (note that Azizi et al. further utilize Multi-Instance Contrastive Learning (MICLe) that uses multiple images per patient as input -- we used augmented versions of the same image).

---

> ### Author Response · Authors · 2021-11-20
> **Response to Reviewer taLm (Part 1)**
>
> We thank the reviewer for their insightful comments. We understand that your main criticism was the motivation behind the segmentation experiment. We have updated the manuscript to reflect that. Please see our comment to all reviewers above clarifying the contributions of our work and additional experiments. We address your comments below.
>
>
> 1. ***“... It is concluded that ViTs outperform CNNs when pretrained using self-supervision.”***
> As we mention in our comment to all reviewers, there seems to be some confusion about the main message of our work. While we do show that ViTs have a small advantage when trained with self-supervision (in terms of relative gains) this is not our main message. Our main message is that, given prior works, it is not obvious that ViTs would perform competitively with CNNs on medical tasks, but we show that they do. We have updated the introduction and the abstract to make that clearer (please see our updated version and our main post for more details).
>
>
> 2. ***“The authors choose one representation for both CNN and transformer model to study the difference....However, I feel that the authors should have chosen U-Net instead of Resnet family as the representative CNN architecture for segmentation….While the authors give 4 reasons on why they chose DeiT as their transformer, there is no reason why resnet was chosen over U-Net.”***
> We give the reasons that we use ResNets and DeiTs, and we have now added SWIN transformers and InceptionV3 to the selection of CNNs and ViTs. For segmentation, we used DeepLab and not a UNet-like structure because (1) as we state in the Related Work, several prior works have shown that UNet-like ViTs can give SOTA results (2) our motivation for showing segmentation examples was to demonstrate that pre-trained of-the-shelf  classification ViTs can reproduce reasonably good features for segmentation (3) we used a DeepLab head (specifically designed for ResNet models with dilated convolutions) to show that even in this disadvantageous setting, ViTs can still perform on par or better than CNNs. We have updated the manuscript to reflect this.
>
>
> 4. ***“There seems to be a significant difference in performance between DeiT and resnet on the ISIC classification task when compared to the other classification tasks (from Table 1). It would be interesting if the authors could shed some light on this observation.”***
> Thanks for noticing this interesting behaviour. Our interpretation is that, since ViTs seem to benefit more from transfer learning, the closeness of this dataset to ImageNet causes ViTs to perform exceptionally well. However, more work is required to show this.
>
>
> 5. ***“It is shown in Isensee et al... If not, the authors should give a good reason why that comparison is not necessary.”***
> Although this is an interesting paper, and indeed pre- and post-processing is an essential component of the pipeline that is used in realistic settings, we argue that this work is irrelevant to our study. It is not our goal to create the optimal segmentation algorithm. Our goal is to show that transferred ViTs produce reasonably good features that can be further used for segmentation tasks. (Please see the comment above and our main post)
>
>
> 6. ***“For the segmentation experiments… A more thorough analysis would include experimenting with a transformer based decoder. Is there any specific reason why the authors left that out, if yes, it would be nice to see that information in the paper.”***
> As we state in the related work, dedicated segmentation architectures using ViTs exist and perform very well (e.g. Chen et al. (2021a)). It has been established that ViTs make strong encoders for segmentation in medical tasks. Our aim here is to show that without any additional architectural modifications, pre-trained ViTs are able to produce reasonably good features for segmentation. (Please see the comment above and our main post)
>
>
> 7. ***“In the discussion section “Interesting Properties of ViTs”, it would be nice to see some emphasis on proving it from the medical imaging perspective for the “Global+Local features” section.”***
> Our experiment in the Appendix D demonstrates this. It was referred to in the paragraph “Other ablation studies”, but we have added a reference to the discussion as well.

---

> > ### Comment · Reviewer_taLm · 2021-11-29
> > **Response**
> >
> > I thank the reviewers for their efforts put in the reply. I think the reviewers have addressed my main concerns well. I am increasing my score.

---

### Official Review · Reviewer_T9mB · 2021-11-03

**Correctness:** 2
**Technical Novelty And Significance:** 2
**Empirical Novelty And Significance:** 2
**Recommendation:** 5
**Confidence:** 4

**Main Review:**

Strengths:
1The manuscript is clearly written with well-organized sections to present the main contributions of this work. It sheds the initial light on the feasibility of replacing CNNs with the Vision Transformers for the medical imaging analysis tasks, specifically for the 2D medical imaging classification and segmentation.

2 The experimental designs are reasonable and makes sense to me to explore the feasibility of replacing CNNs with the recent trends of Vision Transformers. Datasets and experimental details are clearly explained and sufficiently provided for the producibility.

3 The discussion of including the self-supervised manner into the pre-training bears some value as the medical imaging is always lacking of annotations due to the requirement of professional knowledge and intensive laboring to obtain high-quality label. The self-supervised way can alleviate the lacking of annotation problem.

Weaknesses:
1 Medical imaging are often obtained in 3D volumes, not only limited to 2D images. So experiments should include the 3D volume data as well for the general community, rather than all on 2D images. And the lesion detection is another important task for the medical community, which has not been studied in this work.

2 More analysis and comments are recommended on the performance trending of increasing the number of parameters for ViT (DeiT) in the Figure 3. I disagree with authors' viewpoint that "Both CNNs and ViTs seem to benefit similarly from increased model capacity". In the Figure 3, the DeiT-B models does not outperform DeiT-T in APTOS2019, and it does not outperform DeiT-S on APTOS2019, ISIC2019 and CheXpert (0.1% won't be significant). However, CNNs can give more almost consistent model improvements as the capacity goes up except on the ISIC2019.

3 On the segmentation mask involved with cancer on CSAW-S, the segmentation results of DEEPLAB3-DEIT-S cannot be concluded as better than DEEPLAB3-RESNET50. The implication that ViTs outperform CNNs in this segmentation task cannot be validly drawn from an 0.2% difference with larger variance.

Questions:
1 For the grid search of learning rate, is it done on the validation set?

Minor problems:
1 The n number for Camelyon dataset in Table 1 is not consistent with the descriptions in the text in Page 4.

**Summary Of The Paper:**

This work studies the feasibility of replacing CNNs with ViTs for medical imaging analysis on the task of classification and segmentation. Careful experimental designs are conducted to study the comparison between ViTs and CNNs in the aforementioned two medical imaging tasks. Several research questions are raised and answered from the experiments and results. It sheds light on the feasibility of replacing CNNs by the recent vision trends of using ViTs for medical imaging task. It is shown that ViTs can perform better than their CNN counterparts if pretrained via a self-supervision manner.

**Summary Of The Review:**

Overall, it is a good starting point to explore the vision transformers in the medical imaging task. However, some preliminary conclusions are not well-supported. What's more, analysis on 3D imaging and lesion detection are also missing in this work, which are recommended to cover as they are very common for the medical community. So I would like to recommend a weak reject as of now.

---

> ### Author Response · Authors · 2021-11-20
> **Response to Reviewer T9mB**
>
> We thank the reviewer for their positive comments and encouragement. Please see our comment to all reviewers above clarifying the contributions of our work and additional experiments. We address your comments below.
>
> 1. ***“​​So experiments should include the 3D volume data as well for the general community, rather than all on 2D images. And the lesion detection is another important task for the medical community, which has not been studied in this work.”***
> In Section 3 we justify a selection of 7 representative datasets and tasks across a range of modalities to show an interesting phenomenon. It is true that 3D images are also encountered in medical imaging. However, it is impossible to include all different types of data and tasks. Also, replicating our experiments would be difficult since there is no standard way of moving ImageNet pre-trained models to 3D.
>
>
> 2. ***“More analysis and comments are recommended on the performance trending of increasing the number of parameters for ViT (DeiT) in the Figure 3 the DeiT-B models does not outperform DeiT-T in APTOS2019, and it does not outperform DeiT-S on APTOS2019, ISIC2019 and CheXpert. I disagree with authors' viewpoint that "Both CNNs and ViTs seem to benefit similarly from increased model capacity””***
> We respectfully (mostly) disagree. The exact numbers for Figure 3 are provided in Table 4 of the Appendix. While the results are noisy because we only had compute for 5 runs, the same general trends appear. APTOS is indeed an exception, where DeiT-T performs better than its larger variants. Perhaps the tiny size of APTOS results in diminishing returns for large models. We have updated the manuscript to reflect this.
>
>
> 3. ***“On the segmentation mask involved with cancer on CSAW-S, the segmentation results of DEEPLAB3-DEIT-S cannot be concluded as better than DEEPLAB3-RESNET50.”***
> We agree and we have softened this claim. It now reads as “on par or better” than CNNs. Please see the updated version.
>
>
> 4. ***“The n number for Camelyon dataset in Table 1 is not consistent with the descriptions in the text in Page 4.”***
> There was indeed a typo (an extra 9 in between the numbers) that we have now fixed. Thank you for noticing this.

---

### Official Review · Reviewer_aKib · 2021-11-06

**Correctness:** 2
**Technical Novelty And Significance:** 1
**Empirical Novelty And Significance:** 2
**Recommendation:** 3
**Confidence:** 4

**Main Review:**

Unfortunately, I find that the paper oversells and underanalyzes. As I point out below, the authors mischaracterize a few aspects, mostly about the state and interests of medical imaging, including missing important parts of the field. The core of the paper -- a comparison between the two camps -- has very peculiar choices which I believe are inappropriate (not representative of the field or interest), leading to misleading conclusions. Given these points and the results presented, I believe the conclusions are inappropriate. For a paper whose core is experimentation and analysis of that experimentation (rather than, for example, providing a new method or theory), the details of those experiments and the analysis are especially important, (since this is where the proposed insight may come) and in this paper I believe they are not appropriate. I provide suggestions for what might be more appropriate below.

In addition, I think ICLR is not the most fitting venue for this sort of medical imaging experimentation paper, but that may well be debatable and I am happy to be convinced otherwise. However, I believe the authors would have a much bigger impact with such analyses at CVPR or MICCAI.

Overall, I don't believe the paper is ready for publication and ICLR. However, I strongly encourage the authors to continue the work, but importantly:

 -  I encourage them to focus on one specific application, e.g. one of classification or segmentation. Trying to do both leads to the authors cross-motivating (e.g. making segmentation decisions motivated by conclusions in the classification domains) and confusing the insights

 - I encourage them to more faithfully represent the results. I believe this paper in its current form is not on whether ViTs should replace CNNs in medical imaging, but rather on an analysis of ResNet with ViT on 2D medical imaging classification and segmentation.

- Whether ViTs should replace CNNs is a factor of many many aspects, not just accuracy on a few choices. I would encourage the authors to focus more on the fact that they can do a timely and interesting analysis in the future, rather than suggest what the entire field should do based on this narrow analysis.

- I would encourage the authors to focus more on *why* methods perform the way they do. For example, there is a claim that is consistently repeated that ViTs can build larger-space connections and that's why they have a benefit -- but this is simply not shown and theoretical. In practice, these large-space connections can be achieved by the dominant medical image architectures (like the U-Net) thanks to the multi-scale operations. While in theory there is a limit to that, I haven't seen a case in practice where it's relevant.




Details:
 - In the motivation, the authors conflate conclusions from classification with conclusions about segmentation, and apply that motivation to both domains. For example, in many medical imaging domains, segmentation does *not* requite many images as example. In fact, as the authors say the datasets are small -- very small, some segmentation datasets have ~20-30 images in them, and the algorithms achieve really good results (see Antonelli et al 2021, medical decathlon). In fact, these algorithms can do really well even with one example (because they see a lot of pixels/voxels) -- see for example one-shot segmentation in Chaitanya IPMI 2019 or Zhao, Balakrishan CVPR2019; several recent papers that measure the number of segmentation examples (or atlases) that are required for training to achieve great results (often under 10); and so on.  There is substantial literature that shows that zero or one (or very few) medical image with tons of augmentation can achieve remarkable results. The claim that a huge amount of data is needed depends on what is being done (e.g. classification needs more data), the variability of the data and task, and so on. This holds true for a lot of medical imaging tasks, like segmentation, registration, super-resolution, modality transfer, etc.

 - Overall in the motivation several such things are overstated and hand-waived. For example, the authors state that pre-training on imagenet is standard in medical images and outperform those trained from scratch, citing Raghu et al, 2019. The citation, however, concludes the opposite: there's a quote from their abstract "A performance evaluation on two large scale medical imaging tasks shows that surprisingly, transfer offers little benefit to performance, and simple, lightweight models can perform comparably to ImageNet architectures.". It's possible the authors meant something else than I am understand, or meant to cite a different paper? Nevertheless, I don't think the claim is representative of the medical imaging field.

 - I would overall encourage the authors to take out overly aggressive claims, like "there is little to prevent ViTs from becoming the dominant architecture for medical images" -- there are a lot of challenges to ViTs (training, data, etc), even if the accuracies were better (which they are, so far, not)

 -  As noted, the authors miss a substantial amount of literature. Please see many many papers in the medical imaging community (MICCAI/IPMI/MIDL/etc), or even CVPR, which tackle few-shot segmentation, data augmentation for segmentation, test-time segmentation, registration with limited data, etc. I give two example papers above, but there are even papers with zero-shot segmentation. These methods show that (1) a ton of data is not needed, contradicting a substantial amount of the motivation, (2) the predominant architectures are not ResNet based, but mostly Unet-like, among other lessons. Perhaps it's possible that the authors are not familiar with tehse conferences as most of their influence seems to come from the ML community (I could be wrong!) -- if so, I encourage them to look at these venues. There is a lot of relevant literature with substantial leslsons.

 - The authors broadly speak about medical imaging, but only show examples on 2D modalities. As can be seen in papers from the aforementioned venues, from MICCAI/other challenges, etc -- a substantial amount (perhaps the vast majority) or work focuses on 3D data (MRI, CT, etc). While it's perfectly fine to focus on 2D data, this should be clarified in the paper, and not broadly claim that the lessons here apply to all of medical imaging.

 - As the authors say, a lot of the work in medical image analysis is on segmentation and similar tasks (registration, super resolution, synthesis, etc), unlike CV where classification is by far the dominant task. Therefore, I encourage the authors to focus on those tasks. Importantly, the authors make choices that are inspired mostly by classification tasks, like using resnet in both classification and the backbone for segmentaiton. However, by far the most popular and fair architecture is the U-Net, -- not a particular variant of it, but unet like architectures. For example, the nnUNet (a unet-derivative that summarizes and applied architectural lessons of the unet) has won a multitude of challenges in the last year from the field (isenee 2020). Similarly, there are a slew of very popular segmentation tasks, like the BRATS challenge (menze 2014) or the Medical Decathelon (Antonelli 2021). Instead, the authors focus on mostly classification and on resnet backbones, and in the segmentation tasks they do not choose among these popular tasks. This seems inappropriate to me for a paper whose goal is to gain general insight of what is appropriate and successful for medical images -- neither the architectures nor the data chosen are representative of that.

 - In choosing among methods, the accuracy of a specific metric is not usually the only aspect leading to a choice. Most often, several metrics are used (e.g. see for segmentation the Brats challenge, which uses quite a few metrics like Dice, HD95, Jaccard, etc), as well as aspects like train time, training difficulty, inference time, etc. Unfortunately, none of these aspects are considered in the paper -- there is only one metric in the segmentation experiments (IoU, which is not often used in medical imaging, where we mostly use Dice for better or worse), and runtimes and such are not discussed. This latter discussion (runtimes) is crucial to the question posed in the paper, and more metrics compared to Dice are useful.

 - While of course the authros need to choose a training strategy, making very broad claims when there is a very specific training strategy seems misleading to me. How sensitive are the methods to these strategies? Surely as researchers use CNNs vs ViTs they will vary these strategies -- would conclusions carry?

 - I believe the main point of the paper is the final results, and I am not sure I understand the conclusions the authors empahsize. In most scenarios, ViTs are inferior -- such as in training from scratch (which the authors do acknowledge) but even with pre-training in classification in many of the datasets. Perhaps I am reading something wrong, but even with pre-training, I see many cases of both ResNet doing better and DeIT doing better. Overall, I am hard pressed to find a conclusion other than 'ResNet and DeIT perform comparably given pre-training'.

- I would encourage the authors to study more about the differences between the two paradigms -- what can one learn? I know that the claim of 'transformers can build connections over large distances' is popular, but I have not seen this to be true in practice. I think if the authors wish to make this claim, they need to compare it to this ability by the U-Net or other hierarchical architectures which absolutely capture large-space dependencies.

- Overall, the authors should re-evaluate their conclusion. As discussed in several aspects here, there is little evidence that ViTs outperform even in these experiments, and I really believe these are not the right expeirments (not the right architectures or data to work with, etc).




**Summary Of The Paper:**

In this paper, the authors compare ResNets with ViTs on 2D medical (mostly classification) imaging datasets. I think this is a very interesting and timely topic since the current SOTA methods in medical imaging are dominated by CNN-based architectures, and the computer vision community (and NLP) are widely exploring transformers. I was quite excited to read this work, looking forward to good analysis and insight from the conclusion.

**Summary Of The Review:**

The first couple of paragraph (before "Detail") of the review above offer a summary.

---

> ### Author Response · Authors · 2021-11-20
> **Response to Reviewer aKib (Part 4)**
>
> 16. ***“I know that the claim of 'transformers can build connections over large distances' is popular, but I have not seen this to be true in practice. I think if the authors wish to make this claim, they need to compare it to this ability by the U-Net or other hierarchical architectures which absolutely capture large-space dependencies.”***
> Please see response 6, above.
>
>
> 17. ***“Overall, the authors should re-evaluate their conclusion. As discussed in several aspects here, there is little evidence that ViTs outperform even in these experiments, and I really believe these are not the right experiments (not the right architectures or data to work with, etc).”***
> Unfortunately it seems that we have miscommunicated our conclusions as well as our aims. We have updated the motivation and the conclusion of the paper to be clearer. Please see the main response and the updated paper for more details.

---

> ### Author Response · Authors · 2021-11-20
> **Response to Reviewer aKib (Part 3)**
>
> 10. ***“As noted, the authors miss a substantial amount of literature… which tackle few-shot segmentation, data augmentation for segmentation, test-time segmentation, registration with limited data. These methods show that (1) a ton of data is not needed, contradicting a substantial amount of the motivation, (2) the predominant architectures are not ResNet based, but mostly Unet-like, among other lessons.”***
> We respectfully disagree. First, we are well aware of these venues. Our manuscript contains 57 references, far above the mean for ICLR papers, covering CVPR, MICCAI, and MIDL as well as arxiv versions of papers in these conferences. Second, while there is a substantial amount of literature on few-shot learning for CNNs, this is not true for ViTs. The current literature suggests that ImageNet-size datasets are required to perform well. Here we show that this is not the case (please see the main response and the paper for more details). Further, it’s an unfair comparison to say few-shot or zero-shot learning requires little or no data. Nearly all few-shot or zero-shot methods require a lot of data, typically ImageNet, to pretrain the network. In this work we use k-nn classification to show that the features from an ImageNet pre-trained ViT are well separated, as in the case of CNNs -- this can be useful for few-shot algorithms and is indicative of their expected performance.  Finally, we are interested in classification in this work. U-net models are used for segmentation (see related work for variants of U-nets with transformers). For classification ResNets and other CNN-based encoders are used.
>
>
> 11. ***“The authors broadly speak about medical imaging, but only show examples on 2D modalities.”***
> It is true that 3D images are also encountered in medical imaging. However, it is impossible to include all different types of data and tasks. Also, replicating our experiments would be difficult since there is no standard way of moving ImageNet pre-trained models to 3D.
>
>
> 12. ***“Importantly, the authors make choices that are inspired mostly by classification tasks, like using resnet in both classification and the backbone for segmentation. However, by far the most popular and fair architecture is the U-Net, -- not a particular variant of it, but unet like architectures. For example, the nnUNet”***
> Our intention was never to introduce a segmentation model using ViTs that is SOTA. The literature has already shown that ViT-based segmentation algorithms perform well (see related work and main response).
>
> 13. ***“In choosing among methods, the accuracy of a specific metric is not usually the only aspect leading to a choice.”***
> Reporting all metrics is meaningless in many cases due to severe imbalances and the type of the task. We used the appropriate metrics for each dataset following the suggestions from their authors.
>
>
> 14. ***“While of course the authors need to choose a training strategy, making very broad claims when there is a very specific training strategy seems misleading to me. How sensitive are the methods to these strategies? Surely as researchers use CNNs vs ViTs they will vary these strategies -- would conclusions carry?”***
> We motivate the reason that we used these training strategies in Methods. We used (1) random initialized models (2) ImageNet pre-trained models and (3) SSL pre-training. These are training strategies that are commonly used in the literature. Further for a fair comparison between the different training strategies, we selected the learning rate based on the outcome of grid searches. The sensitivity of deep learning models to initialization and hyperparameter selection is a well known problem but we believe that it is out of the scope of our work.
>
>
>
> 15. ***“... I am hard pressed to find a conclusion other than 'ResNet and DeIT perform comparably given pre-training'.”***
> This is exactly the main point of our paper - ViTs perform comparably to CNNs but they require transfer learning (and in some cases SSL) to do so. If put in context with the current literature, this is a non-obvious finding. Please see our main response and the updated version of the introduction for more details.

---

> ### Author Response · Authors · 2021-11-20
> **Response to Reviewer aKib (Part 2)**
>
> 6. ***“There is a claim that is consistently repeated that ViTs can build larger-space connections and that's why they have a benefit -- but this is simply not shown and theoretical. In practice, these large-space connections can be achieved by the dominant medical image architectures (like the U-Net) thanks to the multi-scale operations. While in theory there is a limit to that, I haven't seen a case in practice where it's relevant.”***
> ViTs can build both local and global attention early on - this was previously shown in Dosovitskiyet al. 2020, Raghu et al., 2021 and we show it as well. We refer you to Appendix D, which was referenced in the main text. These plots show the range of the self-attention heads, layer-by-layer. The local+global connections are apparent in early layers, following by highly global features in late layers. Prior works have shown that these local+global connections are crucial for ViTs to perform well, and they require a lot of data to emerge (hence, the use of ImageNet or JFT300M). Comparing to U-Net is a false comparison, it is a segmentation architecture made up of CNN backbones (or ViTs). However, the importance of having both low and high level features at different depths (or scales) has been highlighted both in the U-Net paper (Ronneberger et al., 2015) and in Lin, et al. [1] Given enough data transformers local and global connections emerge in transformers, without the need for feature pyramids or other architectural modifications (which increase the memory and computational cost).
>
>
> 7. ***“In the motivation, the authors conflate conclusions from classification with conclusions about segmentation, and apply that motivation to both domains. For example, in many medical imaging domains. In fact, these algorithms can do really well even with one example (because they see a lot of pixels/voxels) -- see for example one-shot segmentation in Chaitanya IPMI 2019 or Zhao, Balakrishan CVPR2019; … The claim that a huge amount of data is needed depends on what is being done (e.g. classification needs more data), the variability of the data and task, and so on. This holds true for a lot of medical imaging tasks, like segmentation, registration, super-resolution, modality transfer, etc.”***
> This is the same criticism as in point 10, below. Please refer to our response there.
>
>
> 8. ***“For example, the authors state that pre-training on imagenet is standard in medical images and outperform those trained from scratch, citing Raghu et al, 2019. The citation, however, concludes the opposite: there's a quote from their abstract "A performance evaluation on two large scale medical imaging tasks shows that surprisingly, transfer offers little benefit to performance, and simple, lightweight models can perform comparably to ImageNet architectures."***
> The quote you use from the abstract of Raghu et al. is misleading, ironically in a similar manner to our own problems with the title and abstract. If you read the paper in its entirety, you will find that the authors are very careful not to claim that ImageNet pretraining is not broadly beneficial. It is. Just, not as broadly as previously thought. They just found two large datasets (Chexpert and a proprietary diabetic retinopathy dataset) where a large portion of the benefits of transfer learning can be attributed to weight statistics. They are more careful with their wording in the conclusion “Having benchmarked both standard ImageNet architectures and non-standard lightweight models (itself an underexplored question) on two large scale medical tasks, we found that transfer learning offers limited performance gains and much smaller architectures can perform comparably to the standard ImageNet models.”
>
>
> 9. ***“I would overall encourage the authors to take out overly aggressive claims...”***
> We agree. Unfortunately, the tone of our title/abstract/introduction detracted from the real contributions of our work. We have softened the claims you refer to (please see the updated version of the paper). Thank you for pointing that out.
>
> [1] Lin, et al. "Feature pyramid networks for object detection." Proceedings of the IEEE conference on computer vision and pattern recognition. 2017.

---

> ### Author Response · Authors · 2021-11-20
> **Response to Reviewer aKib (Part 1)**
>
> We thank the reviewer for their detailed and thoughtful feedback. Please see our comment to all reviewers above clarifying the contributions of our work and additional experiments. We address your comments below.
>
>
> 1. ***“The core of the paper… has very peculiar choices which I believe are inappropriate (not representative of the field or interest), leading to misleading conclusions.”***
> We respectfully disagree. We motivate the reason we chose these models and experimental settings. We understand that besides our best efforts we did not communicate well some parts of the paper. We have updated the manuscript to reflect that (please see the main response, our comments below, and the updated version of the paper for more details)
>
>
> 2. ***“In addition, I think ICLR is not the most fitting venue for this sort of medical imaging experimentation paper, but that may well be debatable and I am happy to be convinced otherwise”***
> We demonstrate a finding of interest to the community: that ViTs can perform comparably to CNNs (both in a supervised and an unsupervised setting) if transfer learning is utilized, even in distant, from the source, domains and small datasets. This is counter intuitive as ViTs require large datasets to perform well and benefits from transferred representations are expected to diminish as we move away from the source domain. Although our work focuses on medical domains it is a proof of concept that ViTs can work well with limited data (both supervised and self-supervised), which makes it interesting for the ICLR community. Please see the response to all reviewers for more details. Furthermore, the call for papers clearly states that ICLR (https://iclr.cc/Conferences/2022/CallForPapers)  calls for “as well as applications in vision, speech recognition, text understanding, robotics, health care, sustainability, music, games, computational biology, and others.” The list of relevant topics includes: unsupervised, semi-supervised, and supervised representation learning and other topics that fit our work.
>
>
> 3. ***“I encourage them to focus on one specific application, e.g. one of classification or segmentation.”***
> In essence, we have already done this. We concentrate on classification. The segmentation ablation was a way to show an interesting property of classifier ViTs - that they are able to produce relevant features for segmentation tasks as well (without architectural modifications and fine-tuning). It has been established that ViTs are good for segmentation, we cover this in our related work (e,g, Chen et al. (2021a)). We are not claiming to be the first to show ViTs work well for segmentation. Please see the main response for more details.
>
>
> 4. ***“I believe this paper in its current form is not on whether ViTs should replace CNNs in medical imaging, but rather on an analysis of ResNet with ViT on 2D medical imaging classification and segmentation… Whether ViTs should replace CNNs is a factor of many many aspects, not just accuracy on a few choices. I would encourage the authors to focus more on the fact that they can do a timely and interesting analysis in the future, rather than suggest what the entire field should do based on this narrow analysis.”***
> This is an unfortunate miscommunication due to poor choice of the word “should” in our title. Indeed, whether one should replace CNNs with ViTs is a factor of many aspects. This is the reason that we demonstrate some of the interesting properties of ViTs in the discussion and consider a variety of tasks. Please see the updated manuscript and our response to all reviewers above where we clarify our position.
>
>
> 5. ***“I would encourage the authors to focus more on why methods perform the way they do.”***
> We, however, this is not the question we investigate in this work. We feel that the surprising fact that ViTs can do so well in this setting is an interesting and self-contained question to address in a conference paper. Your suggestion is an excellent question for a follow-up work.

---

### Official Review · Reviewer_hd46 · 2021-11-09

**Correctness:** 2
**Technical Novelty And Significance:** 1
**Empirical Novelty And Significance:** 2
**Recommendation:** 3
**Confidence:** 4

**Main Review:**

Strengths:
1. This paper is working on an important question on the capacities of ViT and CNN models for medical image tasks
2. The experiments are clearly described
3. The paper is well-written

Weaknesses:
1. This paper does not specify why they want to focus particularly on medical image tasks. As we all know, there are some major differences between medical images and natural images, in terms of the graphic patterns, scenarios, densities of labels, etc.. As a result, replacing CNNs with ViTs will have different impacts on medical and natural image tasks in different ways. We would like to see more information or rational on this.
2.  In order to support the claim that the usages of GPUs of ResNet and BEIT models can be treated as counterparts, the authors should show statistical data about the numbers of parameters, volumes of computations, and the usages of GPUs of the two models.
3. The comparisons were performed only between the ResNet family models and the BEIT family models. However, these evidences are not enough to draw a conclusion for the substitutability of CNNs and ViTs.
4. The methodology of this paper lacks of novelty.


**Summary Of The Paper:**

The paper under review is working on an important question: Could vision transformers (ViTs) outperform CNNs on medical imaging when performing tasks such as classification, detection, and segmentation? The authors employ ResNet family and DEIT family as the counterparts of CNNs and ViTs. Then they implement these methods for both classification and segmentation models, with and without the integrations of transfer learning and self-supervised learning. By comparing the results of these schemes on several medical image datasets, the authors draw a conclusion that vanilla transformers can reliably replace CNNs on medical image tasks with help of transfer learning and self-supervised learning.

**Summary Of The Review:**

See "Main Review" section for more details.

---

> ### Author Response · Authors · 2021-11-20
> **Response to Reviewer hd46**
>
> We thank the reviewer for their feedback. Please see our comment to all reviewers above clarifying the contributions of our work and additional experiments. We address your comments below.
>
>
> 1. ***In Summary: “​​Could vision transformers (ViTs) outperform CNNs on medical imaging when performing tasks such as classification, detection, and segmentation?”***
> This is not the question we work on. Despite positive comments from most of the reviewers about the clarity of our writing, unfortunately it seems the main point of our work has not been communicated well. We have updated the manuscript to make our contributions more clear. Please also see our comment to all reviewers above.
>
>
> 2. ***“This paper does not specify why they want to focus particularly on medical image tasks”***
> Indeed, medical images are different from natural images. Besides the importance of applications in the medical domain - the differences from the natural domain is the reason that we chose this domain as: (1) It has been shown that ViTs require large datasets to perform well (Dosovitskiyet al. (2020), Touvron et al., 2021) which medical datasets do not have (2) transfer learning can compensate for that in some cases in the natural domain (Dosovitskiyet al. (2020), Caron et al., 2021) but for distant domains (like the medical domain) the benefits from transfer learning are expected to be marginal (Raghu et al., 2019, Neyshabur et al., 2021 ). The literature suggests one would expect that ViTs won’t perform well in distant domains with limited data. We have updated the manuscript to reflect that (see also our main post).
>
>
> 3. ***“In order to support the claim that the usages of GPUs of ResNet and BEIT models can be treated as counterparts, the authors should show statistical data about the numbers of parameters, volumes of computations, and the usages of GPUs of the two models.”***
> The number of parameters is provided in Figure 3. The reason that we did not include run-times and GPU utilization is because these numbers are reported in the original papers. However, these can be added to the Appendix with little effort.
>
>
> 4. ***“The comparisons were performed only between the ResNet family models and the DeiT family models. However, these evidences are not enough to draw a conclusion for the substitutability of CNNs and ViTs.”***
> We motivate the selection of these architectures in the Methods section. We stand by this reasoning and fail to see a justified criticism here. However, during the response period we have reported additional experiments with new CNN and ViT architectures (Inception and SWIN). See our post to all reviewers containing the new results.
>
>
> 5. ***“The methodology of this paper lacks of novelty.”***
> Our work demonstrates an interesting finding that has value to the community - ViTs can perform comparably to CNNs if transfer learning is utilized even if the dataset is small and visually/semantically distant from the source domain. This is a non-obvious finding because the current literature indicates that ViTs require large datasets to perform well, and benefits from transfer learning are expected to diminish as we move away from the source domain. Whether the methodology to prove these findings is novel or not should have little bearing -- in fact, reusing existing methodologies should be preferred as it allows for better context and comparison to prior works.

---

### Author Response · Authors · 2021-11-20
**Response to all reviewers**

We thank the reviewers for their time and thoughtful feedback. Despite positive comments about the clarity of our writing, unfortunately it seems the main point of our work has not been communicated well. We believe that some poor choices in how we introduced and motivated our work led to misinterpretations of our aims and our findings.

To clarify: until now it was unclear whether ViTs can compete with CNNs off-the-shelf on medical tasks. In fact, indications from prior works point suggest that ViTs would struggle to perform in this setting where data size is small and the distance to the source domain is large. But we demonstrate that they can act as drop-in replacements when transfer learning or self supervision is employed. This is not obvious since (1) ViTs are known to rely on large datasets to perform comparably (or better) than CNNs (Dosovitskiyet al. 2020, Touvron et al., 2021) (2) works have shown that the benefits from transfer learning diminish when distant domains are used (e.g. Azizpour et al., 2016) while Raghu et al., 2019 and Neyshabur et al., 2021 show that transfer learning from ImageNet to the medical domain is not as useful as previously thought for CNNs (3) self-supervised learning with ViTs has only been demonstrated to work well with large datasets (Caron et al., 2021).

Our main contribution is to demonstrate this principle, that ViTs can perform comparably to CNNs (both in a supervised and an self-supervised setting) in the medical domain where data is limited and, in the case where transfer learning is used, the distance to the source domain is large. In retrospect, our choice of the word “Should” in the title was perhaps poor as it leads one to expect us to argue that ViTs are unilaterally better than CNNs. Some of the wording in the abstract and introduction may have reinforced this perception. This was not our intent, we rather intended to invite readers to explore the use of ViTs in the medical domain as they have many other compelling properties besides performance.

Further, it was never our motivation with the segmentation experiments to introduce a new SOTA architecture for segmentation. As the reviewers point out, that would require a more rigorous validation. As we discussed in the Related Work, numerous works have shown ViT segmentation architectures such as UNet-inspired models can produce impressive results (e.g. Chen et al. (2021a), Chang et al. (2021) and Hatamizadeh et al. (2021)). Rather, the intention of our segmentation experiments is to show that pre-trained off-the-shelf classifier ViTs have learned good features for segmentation without any architectural modifications. This is interesting because it demonstrates that ViT features can be readily reused for other related tasks. We used a DeepLab head (specifically designed for ResNet models with dilated convolutions) to show that even in this disadvantageous setting, ViT features can perform on par with CNNs.

Regarding our choice of models, we provided a strong motivation for the selection of DeiT and ResNet as representative model families in the Methods section, but several reviewers argued that more models are needed.  Therefore, we provide the results for additional architectures InceptionV3 (CNN) SWIN (ViT) below.


Using random initialization:
 -  | APTOS2019 | DDSM| ISIC2019 | CheXpert| Camelyon
--- | --- | --- | --- |--- |---
**Inception**  | 0.834 $\pm$ 0.012 | 0.922 $\pm$ 0.003 | 0.664 $\pm$ 0.008 | 0.794 $\pm$ 0.001 | 0.953 $\pm$ 0.006
**SWIN-T**    | 0.679 $\pm$ 0.022 | 0.898 $\pm$ 0.005 | 0.619 $\pm$ 0.080 | 0.780 $\pm$ 0.001 | 0.936 $\pm$ 0.002

Using ImageNet pretrained models:
 -  | APTOS2019 | DDSM| ISIC2019 | CheXpert| Camelyon
--- | --- | --- | --- |--- |---
**Inception**   | 0.876 $\pm$0.007 | 0.943 $\pm$ 0.010  | 0.760 $\pm$ 0.011 | 0.797 $\pm$ 0.001 | 0.958 $\pm$ 0.004
**SWIN-T**     | 0.904 $\pm$ 0.005 | 0.965 $\pm$ 0.007  | 0.832 $\pm$ 0.008 | 0.805 $\pm$ 0.001 | 0.968 $\pm$ 0.006


We have updated the manuscript to clarify our aims and findings. The DINO training for both Inception and SWIN-T cannot be completed before the deadline on our hardware, so we will post them in the comments when they finish. From our preliminary experiments (see table above) we confirm that the trends we report on the paper hold with these new models (i.e. that ViTs can perform comparably to CNNs when using ImgeNet pre-training but they perform poorly when training from scratch).

We invite the reviewers to look at our work again. We believe that we have addressed their concerns and clarified our motivation and the novelty in the revision. Our work highlights a finding of practical importance to the community that was not obvious according to the current literature.

---

### Decision · Program_Chairs · 2022-01-20

**Decision:**

Reject

**Comment:**

I recommend a rejection of this paper.

My overall impression is that this is genuinely an interesting topic and this a good basis for a solid paper, however, as pointed by several reviewers, there are multiple unanswered questions due to a very large scope of this work. It might be that a format of a conference paper is not the most appropriate for this work. The authors should consider instead submitting to some of the leading journals on medical image analysis, e.g. IEEE Transactions on Medical Imaging or Medical Image Analysis. I expect this work, as it is mostly empirical, would be appreciated there and could in fact make a much bigger impact if published there.